

# Albedo-Ice regression method for determining ice water content of Polar Mesospheric Clouds using ultraviolet observations from space

Gary E. Thomas[1], Jerry Lumpe[2], Charles Bardeen[3] and Cora E. Randall[1,4]

[1]Laboratory for Atmospheric and Space Physics, University of Colorado Boulder, Boulder, Colorado, USA

[2]Computational Physics, Inc., Boulder, Colorado, USA

[3]National Center for Atmospheric Research, Boulder, Colorado, USA

[4]Department of Atmospheric and Oceanic Sciences, University of Colorado Boulder, Colorado, USA

*Correspondence to*: Gary E. Thomas (thomas@lasp.colorado.edu)





**Abstract.** High spatial-resolution images of Polar Mesospheric Clouds (PMC) from a
camera array onboard the Aeronomy of Ice in the Mesosphere Satellite have been
obtained since 2007. The Cloud Imaging and Particle Size Experiment (CIPS) detects
scattered ultraviolet (UV) radiance at a variety of scattering angles, allowing the
scattering phase function to be measured for every image pixel. With well-established
scattering theory, the mean particle size and ice water content (IWC) are derived. In
the nominal mode of operation, approximately seven scattering angles are measured
per cloud pixel. However, because of a change in the orbital geometry in 2016, a new
mode of operation was implemented such that one, or at most two, scattering angles
per pixel are now available. Thus particle size and IWC can no longer be derived from
the standard CIPS algorithm. The Albedo-Ice Regression (AIR) method was devised to
overcome this obstacle. Using data from both a microphysical model and from CIPS in
its normal mode, we show that the AIR method provides sufficiently accurate average
IWC so that PMC IWC can be retrieved from CIPS data into the future, even when
albedo is not measured at multiple scattering angles. We also show from the model
that 265nm UV scattering is sensitive only to ice particle sizes greater than about 20-
25 nm in (effective) radius, and that the operational CIPS algorithm has an average
error in retrieving IWC of -13$\pm$17%.

## 1 Introduction

Polar Mesospheric Clouds (known as noctilucent clouds in the ground-based
literature) have been studied for over a century from high-latitude ground
observations, but only since the space age have we understood their physical nature,
as water-ice particles occurring in the extremely cold summertime mesopause region.
Their seasonal and latitudinal variations have now been well documented (DeLand et
al., 2006). Interest in these clouds 'at the edge of space' has been stimulated by
suggestions that they are sensitive to global change in the mesosphere (Thomas et al.,
1989). This expectation has been supported recently by a time series analysis of Solar
Backscattered Ultraviolet measurements of PMC (Hervig et al., 2016) and by model
calculations (Lübken et al., 2018).

The Aeronomy of Ice in the Mesosphere satellite (AIM) (Russell et al., 2009) was
designed to provide a deeper understanding of the basic processes affecting PMC,
through remote sensing of both the clouds and their physical environment
(temperature, water vapor, and meteor smoke density, among other constituents).
One of the two active experiments on board AIM is a camera array, the Cloud Imaging
and Particle Size (CIPS) experiment, which provides high spatial resolution images of
PMC (McClintock et al., 2009). CIPS measures scattered ultraviolet (UV) sunlight in the
nadir in a spectral region centered at 265 nm, where ozone absorption allows the
optically-thin ice particles to be visible above the Rayleigh scattering background



issuing from the ~50-km region (Rusch et al., 2009; Bailey et al., 2009). Because of its
wide field of view and 43-second image cadence, CIPS views a cloud element multiple
times in its sun-synchronous orbital passage over the polar region, thus providing
consecutive measurements of the same location at multiple (typically seven)
scattering angles (SA). Together with scattering theory, the brightness of the cloud
(albedo) at multiple angles provides constraints needed to estimate the mean ice
particle size (Lumpe et al., 2013). From the particle size and albedo measurements,
the ice water content is calculated for each cloud element (7.5 x 7.5 km$^2$ in the most
recent CIPS retrieval algorithm). However, over time, the AIM orbit plane has drifted
from its nominal noon-midnight orientation to the point where the satellite is
currently operating in a terminator orbit. Responding to this altered geometry and the
desire to broaden the scope of AIM, new measurement sequences were implemented
to provide observations of the entire sunlit hemisphere, rather than just the
summertime high-latitude region. Because the total number of images per orbit is
fixed by data storage limitations, a new mode (the 'continuous imaging mode') of
observations, with a reduced three-minute image cadence, was implemented in
February 2016. The present sampling in a single Level 2 pixel contains many fewer
scattering angles (often only one). To maintain consistency in the study of inter-
annual variations of PMC, this necessitates a revised method of retrieving ice water
content (IWC) where only a single albedo measurement is available. IWC is a valuable
measure of the physical properties of PMC since it largely removes the effects of
scattering-angle geometry, is a convenient PMC climate variable when averaged over
season, and can be used in comparing with contemporaneous measurements of PMC
that use different observational techniques.
The Albedo-Ice Regression (AIR) method was developed to fill the need to retrieve
PMC IWC with only a single measurement of albedo. Based on the simple notions that
both albedo and IWC depend linearly upon the ice-particle column density, multiple
linear relationships are established between IWC and cloud directional albedo,
depending upon scattering angle. The regressions are derived from three data
sources: (1) the Specified Dynamics version of the Whole Atmosphere Community
Climate Model (SD-WACCM) combined with the Community Aerosol and Radiation
Model for Atmospheres (CARMA); (2) CIPS data for the years 2007-2013, when
multiple scattering angles were available to derive IWC; and (3) Solar Occultation For
Ice Experiment (SOFIE), which provides IWC and particle sizes. These three sources
provide many thousands of albedo-IWC-particle size combinations, from which the
AIR regressions are derived. Although the AIR method may be inaccurate for a single
retrieval of IWC, averages over many observations result in close agreement as the
number of data points increases. The utility of AIR thus depends upon the availability


of large data sets that apply to roughly the same atmospheric conditions. For example,
we will show CIPS results for July and January averages for ascending and descending
portions of the orbit.
In this paper we first describe the theoretical framework relating the scattered
radiance to mesospheric ice particles. It is desirable to use model data to test the AIR
method, without the complications of cloud heterogeneity and viewing geometry. We
utilized a first-principles microphysical model that accurately simulates large numbers
of cloud properties (number density and particle size distribution). The processes
treated by the model include meteor 'smoke' nucleation, growth, and sedimentation,
occurring in a saturated environment at density and temperature conditions provided
by the main global climate model (Bardeen et al., 2010). Several runs for one-day and
multiple-day periods during summer solstice conditions for solar conditions applying
to 1995 were analyzed. Cloud radiances (albedos) at 265 nm were calculated for the
SA range encountered by the CIPS experiment. We chose a set of cloud simulations to
derive a single set of two AIR coefficients through linear regression. The accuracy of
the AIR approximation was then tested on the same data, and on other model runs,
using averages as a function of SA, and increasing IWC threshold values. Thresholding
is necessary to account for the fact that different measurement techniques have
different detection sensitivities. This is not a signal/noise issue, rather the ability to
discriminate PMC against a background that is usually larger than the PMC signal
itself. We show in particular how seasonal means of IWC can be derived from Solar
Backscatter Ultraviolet Spectrometer (SBUV) radiance data, without the need to
derive particle size.
Having tested the technique for model data, we use the same approach with real-life
PMC data collected from CIPS in the normal pre-2016 operating mode. This mode
provided scattering angles needed to define an ice scattering phase function, from
which mean particle size was derived based on assumed properties of the underlying
size distribution (Lumpe et al., 2013). The regressions were run for a period of 40 days
in each of four seasons, each comprising millions of separate cloud measurements,
and from both summertime hemispheres. The results were combined into a single set
of AIR coefficients, and again the AIR technique was tested on monthly averages.
These averages were constructed over all years of nominal spacecraft operations
(2007-2013 in the northern hemisphere, and 2007-8 through 2013-14 in the southern
hemisphere). Note that testing the accuracy of the AIR technique during the nominal
mission period allows the method to be used even during the 'continuous imaging
mode' of CIPS operation.
We then employed highly-accurate data from SOFIE for ice column density and mean
particle size. Since the SOFIE technique uses near-IR solar extinction, it is necessary to





derive scattered radiances from the same algorithm used by CIPS. This exercise was
performed primarily to test whether the derived AIR results are broadly consistent
with those derived from the model and CIPS.
After describing the AIR method, we discuss briefly the application of the method to
a third contemporaneous experiment, the SBUV satellite experiment, which has in
common the same limitations as CIPS in its continuous-imaging mode, namely that
measurements of nadir albedo are made at a single scattering angle. This has already
resulted in a publication (DeLand and Thomas, 2015) where we provided a time series
of PMC IWC from the AIR method extending back to the first SBUV experiment in
124  1979.

**2 Theoretical Basis**
Here we provide a brief overview of the theoretical basis of the IWC retrieval
technique, referring to previous publications for more detail (Thomas and McKay,
1985; Rusch et al., 2009; Bailey et al., 2009, Lumpe et al., 2013). The basic
measurement is PMC cloud radiance $I(\Phi, \theta)$ where $\Phi$ is the scattering angle (angle
between the sun and observation vectors) and $\theta$ is the view angle, which is the angle
subtended by the nadir and observation direction, measured from the point of
scattering. Since the ice layer is optically thin, and secondary scattering is negligible,
the albedo is described by first-order scattering. The ratio of scattered (detected)
radiance to the incoming solar irradiance $F_\lambda$ is the albedo $A_\lambda$, where

$$A_\lambda(\Phi,\theta) = I_\lambda(\Phi,\theta)/F_\lambda = \sec\theta \int_{z_b}^{z_t} dz' \int_{r_{min}}^{r_{max}} dr' \sigma_\lambda(r,\Phi) n(r',z') \tag{1}$$

Here $z'$ and $r'$ are the height and particle radius variables, and $z_b$ and $z_t$ define the
height limits of the ice layer, with the majority of the integrand extending between 83
and 85 km. $r_{min}$ and $r_{max}$ are particle radii which span the particle size regime
responsible for scattering (from ~20 nm to ~150 nm). $\sigma_\lambda$ is the monochromatic
scattering cross-section (cm$^2$-sr$^{-1}$) at wavelength $\lambda$ and scattering angle $\Phi$.
$n(r',z')dr'dz'$ is the number density of ice particles (cm$^{-2}$) in the ranges $r',r'+dr'$ and
$z',z'+dz'$. For CIPS measurements, each camera has a finite bandpass, centered at 265
nm, and is characterized by a function $R_\lambda$ with an effective width of 10 nm
(McClintock et al., 2009). The albedo derived from this instrument is given by

$$A_\lambda(\Phi,\theta) = \sec\theta \int d\lambda' R_{\lambda'} \int_{z_t}^{z_b} dz' \int_{r_{max}}^{r_{min}} dr' \sigma_\lambda(r,\Phi) n(r',z') \tag{2}$$





In the model, the ice particles are assumed spherical, but the scattering theory should
take account of the non-spherical nature of ice crystals. The best agreement of theory
with near-IR mesospheric ice extinction occurs for a randomly rotating oblate-
spheroid shape, of axial ratio two (Hervig and Gordley, 2010). This shape is assumed
in the calculation of $\sigma_\lambda$, which is accomplished through a generalization of Mie-Debye
scattering theory, the T-matrix method (Mishchenko and Travis, 1998). The radius in
the T-matrix approach is defined as the radius of the volume-equivalent sphere. In the
model calculations, we will ignore the view angle effect. In the reported CIPS data,
the $\sec\theta$ factor is applied to the reported albedos, so that $A$ always refers to the nadir
albedo ($\theta = 0$).
The ice water content (IWC) is the integrated mass of ice particles over a vertical
column through the ice layer. Its definition is

$$IWC = \rho \int_{z_b}^{z_t} dz' \int_{r_{min}}^{r_{max}} dr'(4\pi/3)r'^3 n(r',z') \tag{3}$$

$\rho$ denotes the density of water-ice at low temperature (0.92 g-cm$^{-3}$). Anticipating the
results of this study that IWC is linearly related to the column density of ice particles,
$N = \int dr' \int n(r,z')dz'$, we explore the physical basis of this result. As first pointed out
by Englert and Stevens (2007), such a relationship exists for certain SA values, for
which $\sigma \sim r^3$, in which case it is easily seen that Eq. (2) is proportional to IWC.
However, we find that a linear approximation is valid for a much wider range of
scattering angles. To understand this result, we imagine that all particles have the
same radius, so that $n = n_c \delta(r - r_c)$, where $\delta$ is the Dirac $\delta$-function. Then Eqs. (1)
and (3) 'collapse' to simpler results,

$$A_\lambda(\Phi,0) = \sigma_\lambda(r_c,\Phi)N(r_c), \quad IWC(r_c) = \rho V(r_c)N(r_c) \tag{4}$$

Here $N(r_c) = n_c \Delta z$ where $\Delta z$ is the effective vertical thickness of the ice layer.
Eliminating the column density $N(r_c)$, the ice water content is written

$$IWC(r_c) = \rho V(r_c)A(\Phi,0)/\sigma_\lambda(r_c,\Phi) \tag{5}$$

$V(r_c)$ denotes the particle volume. Thus in this special case, $IWC(r_c) \sim A_\lambda(\Phi,0)$. A
superposition of the effects of all participating particle sizes will exhibit a similar
proportionality. When $IWC(r)$ is integrated over all $r$, the contributions from each
size are straight lines, each having different intercepts and slopes.
As previously discussed, the value of the AIR method is in evaluating *average* IWC
(denoted by <IWC>) over many albedo observations made at numerous scattering
angles. The accuracy of the method should be assessed on this basis, not on how well
an individual albedo measurement yields the correct value of IWC. An important issue
is the differing detection thresholds for IWC among the various experiments. In the
case of the scattered-light experiments, the detection threshold depends upon how
well the cloud radiance data can be separated from the bright Rayleigh-scattered
background. The CIPS experiment retrieval method relies upon high spatial resolution
over a large field of view, and the differing scattering-angle dependence of PMC and
the Rayleigh-scattering background (Lumpe et al., 2013). The SBUV retrieval relies
upon differing wavelength-dependence of PMC and background, but primarily on the
PMC radiance residuals being higher (2 sigma) than fluctuations from a smoothly-
varying sky background (Thomas et al., 1991; DeLand and Thomas, 2015). The AIM
SOFIE method is very different, being a near-IR solar extinction measurement in
multiple wavelength bands. SOFIE can detect much weaker clouds than either CIPS or
SBUV. Particle radii values as small as 10 nm can be retrieved from the SOFIE data
(Hervig et al., 2009). To compare the various experiments, it is necessary to 'threshold'
the data from more sensitive experiments with a cutoff value of IWC.
In the next three sections, we present the AIR results from the model, CIPS and SOFIE,
using averages over many cloud occurrences. It is not our intention to compare the
different thresholded data sets to one another (this task will be relegated to a
separate publication), but to illustrate how even measurements made at a single
scattering angle (e.g., SBUV) can yield averaged IWC values that are sufficiently
accurate to assess variations in daily and seasonal averages. These variations are of
crucial value to determining solar cycle and long-term trends in the atmospheric
variables (mainly temperature and water vapor) that control ice properties in the cold
summertime PMC region. We examine the accuracy of AIR through simulations of
scattered radiance from the model, and from CIPS and SOFIE data. Since these data
sources yield particle radii, they can provide both the 'actual' and approximate values
of IWC from the regression formulas. Hervig and Stevens (2014) used the spectral
content of the SBUV data to provide limited information on particle size. Together
with the albedos themselves, they used this information to derive seasonally-
averaged ice water content. They showed that the variation of mean particle size over
the 1979-2013 time period was relatively low (standard deviation of $\pm 1$ nm). They
also found a very small systematic increase with time, as discussed in Sect. 3.
**2. 1 Model Results**
Using a microphysical model as a reference source of IWC 'data' is useful, in the
following ways: (1) in contrast to the CIPS and SOFIE retrieval algorithms, no artificial



assumptions are needed concerning the size distribution of ice particles; (2) radiance
and IWC may be calculated accurately, so that effects of cloud inhomogeneity are
absent; and (3) limitations due to background removal are absent. In addition, to gain
insight into the accuracy of the AIR approach, it is sufficient to work with
monochromatic radiance at the central wavelength of the various passbands. The
integrations of Eqs. (1) and (3) were approximated by sums over 1-nm increments of
radius, and over all sub-layers within the model ice cloud (a typical ice layer is several
km thick.). The model height grid is variable, being highest in the saturated region
where the smallest layer thickness is 0.26 km (see Bardeen et al., 2010 for more
details). We then performed the linear regression for SA values over which CIPS
observations are made.
Figure 1 displays the regressions for six scattering angles, and 2514 individual model
clouds. The units of IWC are g-km$^{-2}$, or μg-m$^{-2}$, which are commonly used in the
literature. Each plot is divided into two groups according to the effective radii $r_{eff}$ for
each cloud. $r_{eff}$ is defined in the literature (Hansen and Travis (1974) as

$$r_{eff} = \int dr' n(r') r'^3 / \int dr' n(r') r'^2 \qquad (6)$$

Figure 1 clearly illustrates that particle size contributes to the 'scatter' from the linear
fits. It also shows the existence of a non-zero intercept of IWC vs albedo. The non-zero
intercept was at first surprising since we expected that for an albedo of zero, IWC
should also be zero. In fact, we found that the linear relationship breaks down for very
small albedo, and the points in the plot narrow down as $A_\lambda \to 0$ (not shown). In albedo
units of $10^{-6}$ sr$^{-1}$ (hereafter referred to as 1 G) this departure from linearity occurs for
A<1 G and IWC<10 g/km$^{-2}$, conditions which fortunately are below the sensitivity
threshold of CIPS and SBUV, and therefore unimportant for our purposes. For more
sensitive detection techniques, this limitation must be kept in mind. A limitation of
the present model (not necessarily all models) is that it does not simulate the largest
particles in PMC and the largest values of IWC, as seen in both AIM SOFIE and CIPS
experiments. The largest model IWC value is 180 g-km$^{-2}$ and the largest effective
radius is 66 nm, whereas CIPS and SOFIE find particle radii up to 100 nm and IWC up
to 300 g-km$^{-2}$. This limitation is irrelevant for the AIR CIPS results (to be discussed),
but could limit the application of the AIR technique to SBUV data. In Sect. 3 we will
return to the issue of the AIR accuracy, as applied to SBUV data.
We chose to use averages for the entire model run, which includes different latitudes,
longitudes, and UT, but the data can be subset in many different ways. It is certainly
preferable in data sets to choose a small time and space interval over which
temperature and water vapor are not expected to vary, but this is not necessary for



the model. All that we ask of the model is whether the AIR results provide an accurate
estimate of <IWC>, taken over the ensemble of model cloud albedos calculated at a
variety of scattering angles.
As discussed above, we are also interested in the accuracy of AIR in the thresholded
data, that is, how AIR represents <IWC> in comparisons of data sets with varying
detection sensitivities to PMC. Figure 2 displays the error in the ensemble-average
(2488 model clouds) as a function of the IWC threshold and scattering angle. Despite
the large data scatter from the linear fit shown in Fig. 1, the averaging removes almost
all the influence of the 'random error'. In this case, the overall error is less than 3%.
The influence of particle size is of course not a random error, but acts like one in the
averaging process. However, the AIR coefficients also depend weakly upon the mean
effective radius, defined in Eq. (6) for a single cloud, which varies from one latitude to
another and from year to year. The effect of variable $r_{eff}$ on the AIR error is discussed
in Section 3.
**2.2 AIR Results from CIPS**
A detailed description of the Version 4.20 CIPS algorithm, together with an error
analysis of individual cloud observations, was presented in Lumpe et al. (2013). Here
we describe only what is necessary to understand how IWC is derived from the data.
Even though an accurate determination of the scattering-angle dependence of
radiance (often called the scattering phase function) is obtained by seven
independent measurements, this does not fully define the distribution of particle
sizes. Instead, additional constraints need to be introduced to derive the mean
particle size. The particles are assumed to be the same oblate-spheroidal shape as
defined for the model calculations, and to have a Gaussian size distribution (see eq.
11 in Rapp and Thomas (2006). A relationship between the Gaussian width *s* and the
mean particle radius $r_m$ is derived from a relationship found in vertically-integrated
lidar data (Baumgarten et al., 2010). The net result is that two parameters, the mean
particle size and the Gaussian width, are retrieved from a given scattering phase
function. However, there is only one independent variable, since the two are related
by $s(r_m)$. Thus Eq. (3) simplifies to

$$IWC = \rho V(r_m) A(\Phi = 90°, 0) / \sigma_\lambda(r_m, \Phi) \qquad (7)$$

*V* denotes the mean ice particle volume evaluated at $r_m$. *A* refers to the retrieved
albedo, corrected to view angle $\theta = 0$ and interpolated to scattering angle $\Phi = 90°$.
Note the resemblance of Eq. (7) to Eq. (5). $A(\Phi = 90°, 0)$, along with $r_m$ and $IWC$
are    products    reported    in    the    CIPS    PMC    data    base,    found    at
(http://lasp.colorado.edu/aim/).  $\sigma_\lambda(r_m, \Phi = 90°)$ is  the  mean  scattering  cross-



section, integrated over the assumed Gaussian distribution with mean radius $r_m$ and
distribution width, $s$.
Before discussing the AIR results, we first apply the CIPS algorithm to the model data
to test how well it works on a set of realistic particle sizes. As mentioned earlier, UV
measurements of ice particles are not sensitive to particle radii < 20-25 nm. We
applied the CIPS algorithm to 6672 model clouds, using seven scattering-angle points,
spanning the range 50°-150° (the results are insensitive to the values chosen). We
then calculated the % difference between the exact model calculation of IWC and the
simulated CIPS retrieved IWC for every model cloud. Figure 3 shows the result as a
function of $A(\Phi = 90^o)$. The mean difference and standard deviation for two model
days is -13$\pm$17%. With the caveat that not all ice is retrieved, only a large subset of
CIPS IWC data have an acceptable accuracy (an average of 84% of the modelled ice
mass is contained in particles with radii exceeding 23 nm). We note that IWC in the
model used to derive the AIR approximation refers to *all* particle sizes.
The procedure for deriving AIR coefficients from the CIPS data is as follows: (1)
Regression coefficients were derived from data pertaining to 0-40 days from summer
solstice (day from solstice, DFS=0 to 40) on every third orbit. This meant that ~200
orbits per season were used. The regression analysis was performed on four years of
data (2010-2013). The data were binned in 5-degree SA bins and only the best quality
pixels with six or more points in the phase function were used; (2) Data from each
northern and southern summer season were treated separately. The coefficients and
standard deviations of the fit were then interpolated to a finer SA grid from 22° to
180° in increments of 1°; (3) The coefficients from each hemisphere were averaged,
and these coefficients were then used to create an AIR IWC data base, to accompany
the normal CIPS products. As previously shown, the AIR data applies to the ice mass
of 'UV-visible' clouds, not to their total ice mass.
We emphasize that using the AIR data is unnecessary for seasons prior to the northern
summer season of 2016 – however the AIR data have great importance since that time
because the observing mode was changed, resulting in measured phase functions that
contain many fewer (and often only one) scattering angles. As illustrated in Fig. 4, it is
trivial to infer both IWC and A(90°) from a single measurement of albedo. This
alternative 90-deg albedo value, ALB_AIR, is now included along with IWC AIR in the
CIPS Level 2 data files. Fig. 5 shows the AIR results for monthly-averaged IWC (July and
January) compared to the same averages of the more accurate results from the
operational (OP) retrieval described in Lumpe et al. (2013). The data have been
separated into different hemispheres, and into ascending and descending nodes of
the sun-synchronous orbit, and apply to the years of the nominal operating mode. The
ALB_AIR results are systematically higher than the operationally retrieved 90-deg





albedo, whereas there is no consistent bias in the IWC (AIR) value compared to the
operational product. However, for both quantities the interannual changes between
the AIR and OP results agree very well. This is reflected in the very high correlation
coefficients of the two sets of values. A more stringent test of the AIR method comes
from daily values of CIPS IWC. Shown in Figs. 6 and 7 are polar projections of IWC (AIR)
and the more accurate operational IWC data product. These 'daily daisies' are taken
from overlapping orbit strips pertaining to 28 June of two different years. Figure 6
shows data from 2012, when CIPS was still in normal mode, and the AIR result shows
remarkable agreement with the operational IWC data. By 2016 (see Figure 7) CIPS is
in continuous imaging mode and the standard IWC retrieval is limited due to the
scarcity of pixels with three or more scattering angles. Here the AIR approach is clearly
superior and does a good job of filling in the polar region where CIPS detects high-
albedo clouds. The differences in patterns are due primarily to variations of particle
size, rather than errors in the AIR method.
AIR accuracy can also be tested in the study of latitudinal variations. Figure 8
compares daily-averaged IWC from the CIPS Level 3C data, for both the standard and
AIR algorithms, for the Northern Hemisphere 2011 season. It is clear that AIR is
adequate even for 24-h averages. For example, it is capable of defining the beginning
and ending of the PMC season, a metric that has valuable scientific value (e.g., Benze
et al., 2012)
**2.2 Results from SOFIE**
A third independent data set of IWC and particle size is available from the AIM SOFIE
experiment. SOFIE provides very accurate values of IWC, through precise near-IR
extinction measurements, independent of particle size. It assumes the same Gaussian
distribution of particle sizes as CIPS, so that the reported value of mean particle radius
$r_m$ is consistently defined. SOFIE data are useful to investigate the extent to which the
AIR approximation can be applied to an independent data set. To do so, it is necessary
to calculate 265-nm albedo at various SA values, given the values of $r_m$, ice column
density $N$ from the data base, and the mean cross-section $\sigma_\lambda(r_m, \Phi)$. The latter
quantity is  averaged over the assumed Gaussian distribution. The equation for the
albedo is

$$A_\lambda(\Phi, 0) = \sigma_\lambda(r_m, \Phi) N \qquad (8)$$

Given $A_\lambda(\Phi, 0)$ and IWC for each PMC measurement (one occultation per orbit), we
can once again perform regressions and find AIR coefficients for the SOFIE data set.
The comparison of AIR results from all three data sets is shown in Fig. 9,  where the





constant term $C$ is the y-intercept and $S$ is the slope in the AIR regression

$$IWC(AIR) = C(\Phi) + S(\Phi) * A(\Phi, 0) \qquad (9)$$

Figure 10 displays the results from the three data sets, expressed as contour plots of
AIR-derived IWC as functions of SA and Albedo. This comparison shows that the three
sets of IWC resemble one another far better than would be anticipated from the AIR
coefficients in Fig. 9, where the constant coefficient differs significantly between data
sets. Since the result of the regression in yielding IWC is more significant than the
coefficients themselves, the comparisons of Fig. 10 are the more appropriate
diagnostic. The fact that the IWC derived from AIR is more accurate than would be
expected from the differing coefficients is due to the fact that the errors of the
constant and slope coefficients are anti-correlated. The agreement between the three
results will be even better when taken over a large data set with variable SA and
albedo. The comparisons of IWC from different satellite experiments as a function of
year and hemisphere will be the subject of a separate publication.
Figure 11 shows that the regressions with AIM SOFIE data obey a linear relationship
between IWC and albedo for IWC <220 g-km$^{-2}$, but for SA values <90°, AIR
overestimates IWC by up to 15%, depending upon the SA. For SA=100° the regressions
are still linear up to 300 g-km$^{-2}$, values above which are seldom encountered in the
data.
**2.3 SBUV data**
The AIR coefficients from the model have been used by DeLand and Thomas (2015) to derive
mean IWC from SBUV data, which spans the largest time interval of any satellite data set
(1979-present). The 273 nm wavelength used in the SBUV Version 3 analysis is sufficiently
close to the effective wavelength of the broader passband of the CIPS cameras (Benze et al.,
2009) that the same coefficients may be applied to both data sets. The accuracy of the average
IWC results was estimated by removing half the data from an entire season and comparing
the two results. For a highly-populated region (more than 1000 clouds per season at latitudes
higher than 70°), the changes in IWC were $\pm 3 - 5$ g-km$^{-2}$. For a less populated region ($50° - $
$64°$ latitude) where there were many fewer clouds (<50), the changes were larger, $\pm 5 - 10$ g-
km$^{-2}$. Even the larger errors are sufficiently small for intercomparison of SBUV and
contemporaneous PMC measurements. Figure 12 shows a comparison of SBUV IWC, using
the model AIR coefficients, to the results of a more accurate determination of IWC derived
from particle size determinations using SBUV spectral information (Hervig and Stevens, 2014).
The comparison is for data residuals from July averages over the time series 1979-2017. Given
the different assumptions underlying the two data sets, the agreement is very good (with an
rms difference of 3% for the residuals, and 5% for the actual values of <IWC>).
**3 Effects of Mean Ice Particle Size**


The AIR approximation is based on the notion that particle size effects can be ignored.
In fact, the particle size (or more accurately, $r^3$) is a principal 'driver' of $<IWC>$
itself, so it is not obvious that particle size effects may be neglected. Column density
also drives IWC, and the dependence of albedo on density adequately captures this
part of the variability (albedo is strictly linear in column density). The AIR slope term
is proportional to $\frac{r^3}{\sigma_\lambda(r,\Phi)}$ averaged over a distribution of particle sizes, $r$. Since
$\sigma_\lambda(r,\Phi) \sim r^{3-5}$ (where the exponent depends upon $\Phi$) then averaging over many
values of $r$ results in a slope term that, in the limit of large number, depends
predominantly on $\Phi$. This is an example of "regression to the mean", and illustrates
how the approximation is designed to work for large numbers of clouds. In a fictitious
case where the mean cloud particle size is larger in one year than another, but the
cloud column number remains the same, the mean albedo would increase according
to Eq. (8), resulting in an increase of <IWC>. We might expect that the slope term
would be different in the two cases. Our study with three different data sets shows
that the regression slope itself remains almost the same among the three data sets,
despite their differing in mean particle size.
In fact, Hervig and Stevens (2014) found from SBUV spectral data a small long-term
trend in <IWC> and in addition a trend in the mean particle size (+0.23 ± 0.16
nm/decade). This contributed an additional 20% to the overall long-term trend in
<IWC>. The ignored dependence on mean particle size using the AIR method thus
adds a systematic uncertainty in derived <IWC> trends, which can be as large as 20%,
according to Hervig and Stevens (2014). This error undoubtedly varies inversely with
the number of clouds in the averaging process. For example, the number of CIPS
observations per PMC season greatly exceeds that of SBUV, so that the error in <IWC>
should be correspondingly smaller.
**4 Conclusions**
We have described the theoretical basis and accuracy for an approximation for retrieving the
average ice water content (IWC) of Polar Mesospheric Clouds (PMC) from measurements of
UV albedo at a single scattering angle. This approach provides a continuous set of consistent
CIPS measurements of IWC from year to year, regardless of the number of scattering angles
for which albedo at a single location is measured. The consistent AIR IWC data base enables
robust IWC comparisons throughout the AIM mission, from 2007 to the present. A
comparison of IWC derived from the microphysical model and from the CIPS algorithm
suggests that CIPS is capable of measuring 84% of the total ice content of PMC (for particle
sizes exceeding ~23 nm). The accuracy for measuring the total (over all particle sizes) IWC is -
$13\pm17\%$. The AIR approximation is less accurate for high IWC (>220 g-km$^{-2}$), but very-high
mass clouds (IWC> 300 g-km$^{-2}$) are infrequent and do not influence seasonal averages of IWC.
The accuracy of the AIR results for ensemble averages has a small systematic dependence on



mean particle size- the error depends inversely on the size of the ensemble. The inter-annual
and hemispheric variations of IWC derived from CIPS and SBUV measurements throughout an
entire 11-year period (2007-2018) will provide detailed information on PMC variability over
the recent solar cycle 24.
**Figure captions**
Figure 1. Linear regressions of model PMC albedo versus model PMC ice water content. The
black points represent model clouds with $r_{eff}$ <40 nm. The red points apply to $r_{eff}$ >40 nm. The
blue line is the linear least-squares fit to all points. (a) through (f) are for different scattering
angles.
Figure 2. Errors of ensemble averages, <IWC> using the AIR approximation, taken over all
cloud model simulations for conditions of summer solstice. <IWC> is 'thresholded' by the
variable IWC in the vertical axis, so that <IWC> applies to all values above IWC.
Figure 3. Differences of IWC derived from the model cloud 'data' and the accurate IWC from
the model, plotted against the 265-nm albedo (in G units, see text), evaluated at SA=90°. The
error bars are the standard deviations in intervals of 2G.
Figure 4. Illustration showing how IWC=98 g-km$^{-2}$ (horizontal arrow) and A(90°) = 16 G (thick
downward arrow) are derived from the AIR method from a single measurement of cloud
albedo at 60 G and SA=40° (upward arrow). Each straight-line plot is calculated from Eq. (9).
Figure 5. Comparison of CIPS A(90°) (top) and <IWC> (bottom) calculated from the operational
(OP) and AIR algorithms. Data points correspond to July northern hemisphere (NH) and
January southern hemisphere (SH) averages in a 5-degree latitude bin centered at 70°. Left
and right panels are for ascending and descending legs data, respectively.
Figure 6. Polar projection map of IWC from CIPS, Day 180 (28 June 2012). Left and rights panels
show the operational IWC product and the AIR result, respectively.
Figure 7. Same as Fig. 6 except for 28 June 2016.
Figure 8. Filled circles and dotted line: IWC (AIR) averaged over 1-deg latitude bins centered
on 70° (green) and 80° (blue), and over 15 orbits (from which daily averages are derived). Solid
line: standard L3C IWC averaged in the same way.
Figure 9. AIR coefficients for three different sources of IWC and particle size: Model (solid line
with open circles), CIPS (solid line), and SOFIE (dashed line).
Figure 10. Contour plots of the AIR approximations for IWC versus cloud albedo (G) for the
three data sources: (a) model, (b) SOFIE, and (c) CIPS.



Figure 11. Examples of SOFIE AIR regressions for two (specified) scattering angles, (a) 80º and
(b) 110º.
Figure 12. Comparison of annually-averaged northern hemisphere July-averaged residuals
(<(IWC>-long-term mean) derived by two independent methods from SBUV 273 nm albedo
data. Black curve: <IWC> derived from the AIR approximation. Blue curve: <IWC> derived from
the same SBUV albedo data, but including mean particle size variations (see text). A three-
year smoothing has also been applied.

### Data Availability

The CIPS operational PMC data, along with the AIR data, can be found at
http://lasp.colorado.edu/aim/).

### Author Contribution

Author G.T. formulated the AIR approximation, and derived the AIR coefficients from the
microphysical model (provided by author C.B.), and from the AIM SOFIE data
(http://sofie.gats-inc.com/sofie/index.php). Authors J.L. and C.R. calculated the AIR
coefficients from the CIPS data.

### Disclaimer

The authors declare that they have no conflict of interest.

### Acknowledgements

We thank M. DeLand and M. Hervig for providing us with the data used in Fig. 12. We
gratefully acknowledge the tremendous effort of the engineering, mission operation and
data systems teams whose dedication and skill resulted in the success of the CIPS
instrument. AIM is funded by NASA's Small Explorers Program under contract NAS5-03132.

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



Figure 1





Figure 2





Figure 3





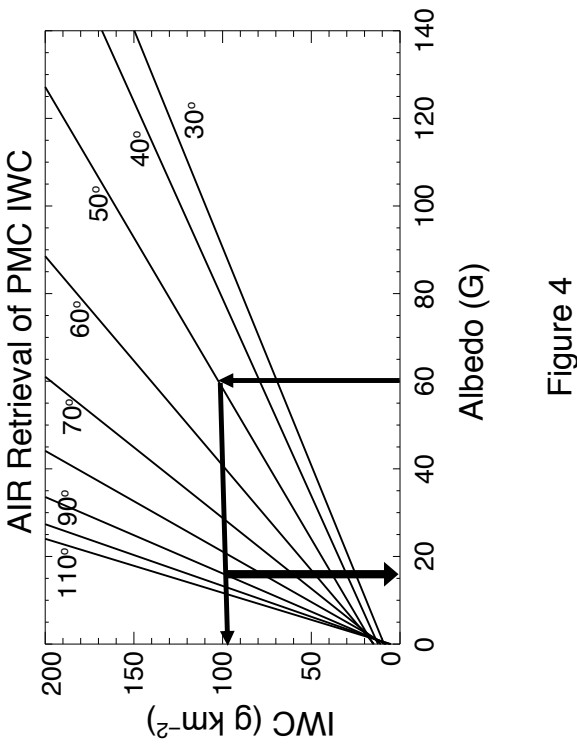

Figure 4





Figure 5


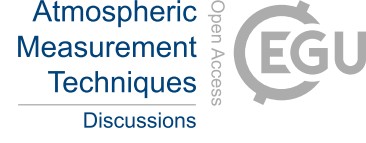

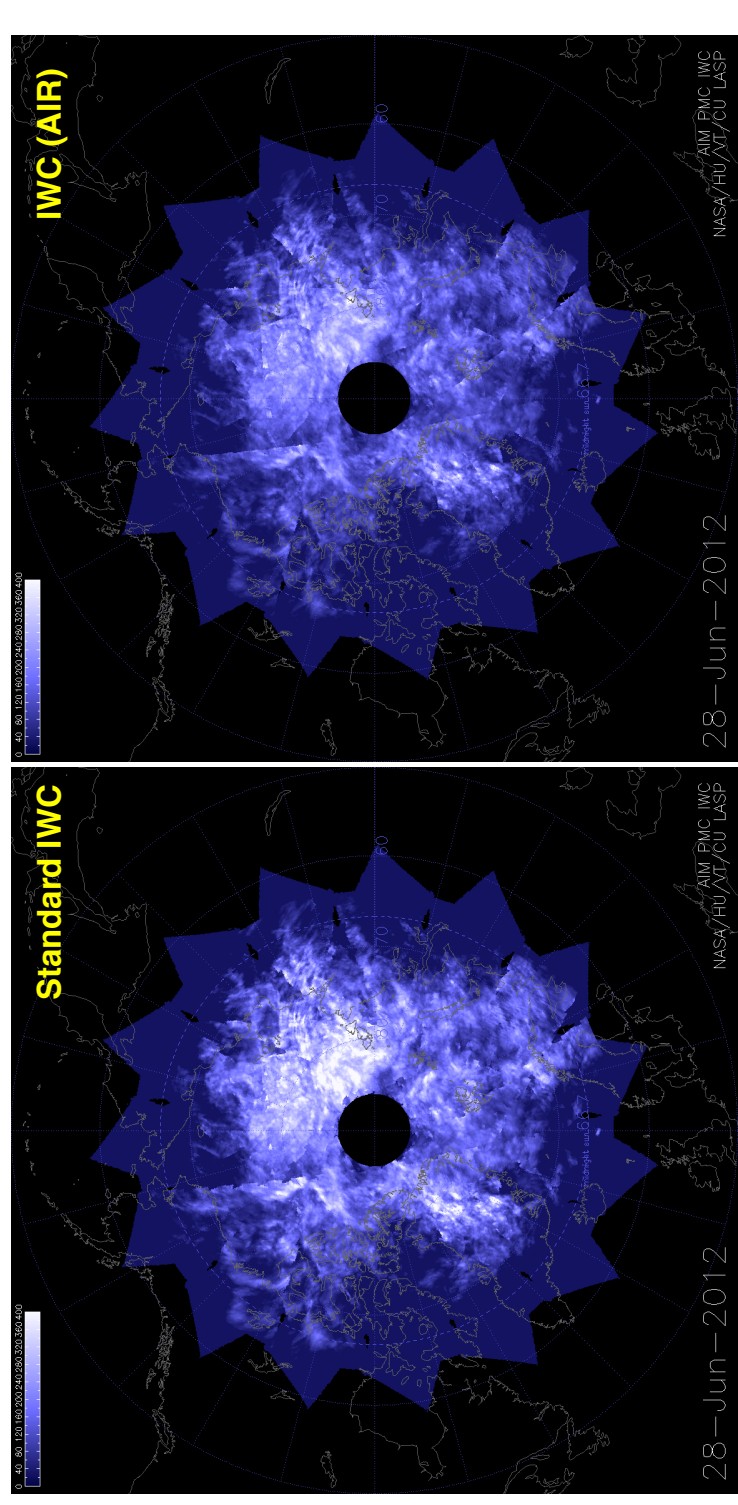

Figure 6



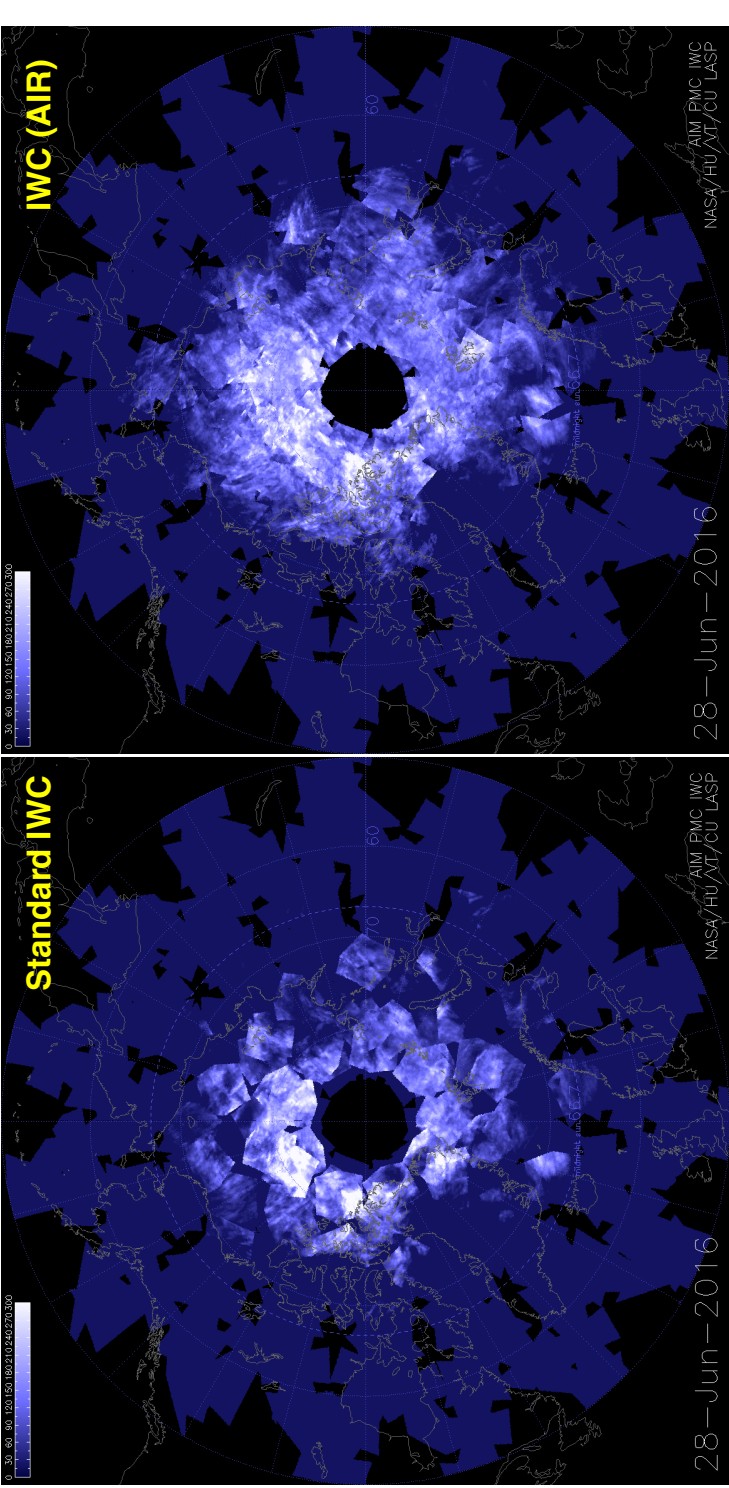

Figure 7







Figure 8



Figure 9


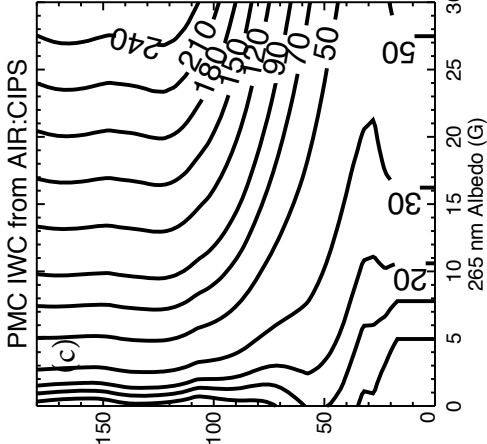

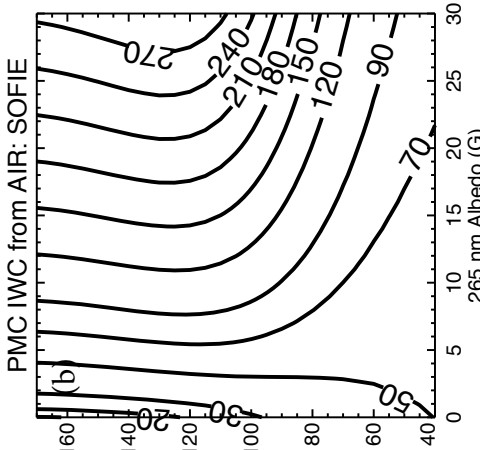

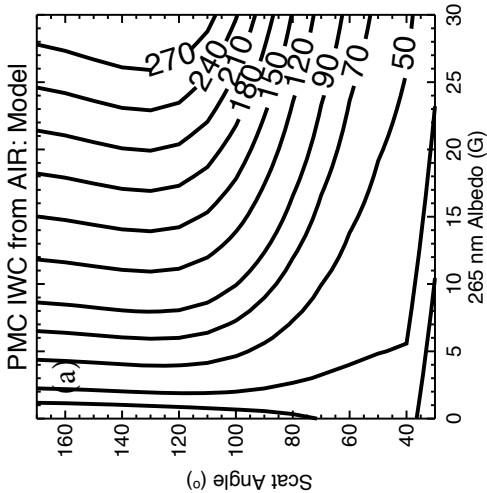

Figure 10



Figure 11

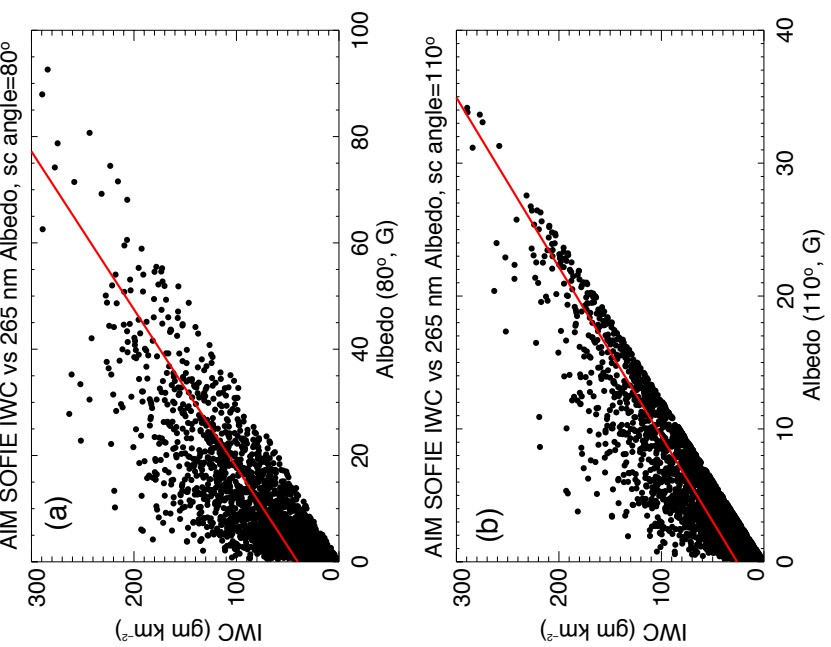



Figure 12