# Peer review of "Albedo-Ice regression method for determining ice water content of"

_Atmospheric Measurement Techniques, 2018_

## Referee Comment (RC1) · Anonymous Referee #1 · 23 Dec 2018

General comments:

This manuscript deals with an alternative approach to retrieve the ice water content (IWC) of PMCs from satellite-borne UV-backscatter observations with the CIPS instrument on the AIM satellite. Due to orbital drifts CIPS is since 2016 operated in a different way and the original approach to retrieve IWC is not possible any more for a large part of the measurements. The novel approach estimates IWC from backscatter measurements at a single scattering angle. Overall, the approach works well. The paper is generally well written and is suitable for publication in AMT. I ask the authors to consider the comments listed below. In my opinion the paper should be accepted, once

the (mainly minor) issues listed below were addressed.

Specific comments:

Line 16: "265 nm" -> "265 nm"

Line 114: "..it is necessary to derive scattered radiances from the same algorithm used by CIPS."

I don't really understand this statement "derive scattered radiances from the same algorithm used by CIPS." CIPS measures scattered radiances and the algorithms are used to infer physical properties of PMCs, right? I guess I'm missing a point?

Line 138: "the monochromatic scattering cross-section"

It actually is the "differential" scattering cross section. Perhaps this can/should be added.

Line 140: "number density of ice particles"

Perhaps better "column density"?

Equation (2): I suggest using a slightly different symbol for the albedo than in equation (1). The left hand sides of equation (1) and (2) are the same, but the right hand sides differ.

Line 167: add space before "denotes"

Line 209: "(2) radiance and IWC may be calculated accurately, so that effects of cloud

inhomogeneity are absent;"

I don't fully understand the meaning of this sentence. Is the microphysical model a 1-D model?

Line 216: "thick.)." -> "thick)."

Same line: I suggest adding "resolution" to read "The model height grid resolution is variable, being highest . . .", otherwise "being highest" doesn't make sense.

Line 220 and Figure 1: I'm wondering how the linear regressions are actually done. Is it a single linear fit of y vs. x, or do you fit both y vs. x and x vs. y and determine an average slope and offset? Alternatively, there are routines taking both differences in x and y direction into account, when minimizing chi-square. There may be a large difference between fitting y vs. x or x vs. y.

Line 227: "In fact, we found that the linear relationship breaks down for very small albedo,"

The reason is probably, that particle populations with really small particles (< 10 nm) have a non-zero IWC, but the albedo is essentially zero, right?

Figure 1: Both the IWC and the albedo values have lower limits. Perhaps I missed it, but what is the reason for this lower limit. For IWC the limit seems to be 20 g / km$^2$.

Figure 2: The caption should clarify that the displayed error is a relative error given in

Line 216: Closing parenthesis missing after "Rapp and Thomas (2006)".

Line 218: "mean DIFFERENTIAL scattering cross-section"?

Line 289: "The mean difference is . . . -13

Looking at the Figure, the mean difference seems to be larger. Also, what is the reason for the characteristic variation of the difference with increasing albedo? Is there a simple explanation?

Line 289: " .. subset .. have" -> ".. subset .. has"

Line 303: "As previously shown, the AIR data applies to the ice mass of 'UV-visible' clouds, not to their total ice mass."

I don't really understand this statement. What is the meaning of "UV-visible clouds"? Was this shown in the current paper? If not, please provide a reference.

Caption Figure 4, line 3: "SA = 40 deg" should read "SA = 50 deg"

Line 366: "AIR overestimates IWC by up to 15

AIR may also underestimate IWC, right? I also think that the overestimation may be much larger than 15

Line 367: "SA = 100 deg" -> "SA = 110 deg"

Line 375: "The accuracy of the average IWC results was estimated by removing half the data from an entire season and comparing the two results. "

I'm not sure, I fully understand this statement. AIR is applied to SBUV data and then you split the data set in half and compare the results. How does this allow you to estimate the accuracy of the "average IWC results"? I guess I'm missing the point here ?

Next sentence: You report on "changes" of $\pm$ 3 - 5 g / km$^2$ etc., but were there any systematic differences?
* * *

---

## Referee Comment (RC2) · Anonymous Referee #2 · 14 Jan 2019

This manuscript describes the retrieval of the ice water content (IWC) of mesospheric clouds based on measuring cloud albedo. This retrieval method has been developed with particular focus on AIM/CIPS since changes in orbit have made the original IWC retrieval based on phase function analysis impossible. However, the method is also applicable to other mesospheric datasets like SBUV. This gives the method potential importance for inter-comparisons of mesospheric datasets and for the analysis of long-term variability. The approach is straight-forward, and the results are convincing. I recommend the manuscript for publication after some minor revisions. I mainly would like to see a number of clarifications.

[Figure]

One issue I would like the authors to discuss more: Why is the AIR method not even better than described in the manuscript? The AIR method is based on finding a linear relationship between ice water content and cloud albedo. For typical mesospheric clouds, one can argue for such a relationship even on a theoretical basis, as long as scattering coefficients depend on the particle size to a power in the vicinity of 3. The authors show that the AIR method works well on a statistical basis. However, the AIR results presented in the manuscript show much scatter in the relationship between albedo and IWC, and the authors point out in several places that we cannot expect AIR to work for individual IWC retrievals. I would like to see more discussion on why the method is "not better than this". A reference is e.g. Hultgren and Gumbel (J. Geophys. Res., 119, 14129-14143, 2014). That paper shows many examples of a close relationship between cloud scattering coefficient and ice mass density, which works well even for the altitude-dependent quantities, not only for the column-integrated quantities considered in the current paper. Can I dare the authors to make a more quantitative statement: Can we take the AIR results for real and apply the method to individual retrievals, by providing a suitable statement about the error bar of such individual AIR IWC retrievals? How large (in percent) would such an error bar be?

Section 2 (Theoretical basis): The major result of this study is that IWC is linearly related to cloud albedo. Therefore, the description in line 156-160 is confusing. In line 156-157, the authors refer to "the results of this study that IWC is linearly related to the column density of ice particles". The fact that IWC is linearly related to the column density of ice particles is somehow trivial (although dependent on the details of the particle population). In line 158-159, it is also stated that "As pointed out by Englert and Stevens (2007), such a relationship exists for certain SA values..." However, the relevant finding by Englert and Stevens is about the relationship between IWC and albedo. I therefore suspect that this paragraph should be about the relationship between IWC and albedo, not between IWC and column density. Please clarify and reformulate.

Section 2.1 (Model results): Please describe in more detailed the processes included

in the model simulations and the resulting variability in mesospheric clouds. Is gravity wave activity included in the simulations? Does the cloud database include multiple layer clouds or other conditions that may lead to clouds deviating from a straight growth/sedimentation scenario?

Section 2.2 (AIR results from CIPS): Figure 8 shows the AIR method applied to the CIPS from the Northern Hemisphere 2011. When it comes to demonstrating that the AIR method works, this choice of season is unfortunate. CIPS data from the years 2010-2013 has been used in the regression analysis to "train" the model. When subsequently investigating the ability of the model to retrieve IWC, a season should be chosen that has not already been used to train the model. I suggest to choose another season.

Some other details:

Line 84: The notation "meteor 'smoke' nucleation" may be misleading. It is better to write "nucleation on meteoric 'smoke'".

Line 274: To avoid confusion, please clarify what is meant by "mean ice particle volume evaluated at rm", i.e. make clear that you refer to an integration over the Gaussian particle size distribution.

Line 287: Clarify that by "simulated CIPS retrieved IWC" you mean the AIR result.

Line 394: The authors refer to n = 3-5 as typical exponents for the size-dependence (sigma $\sim$ r^n) of the scattering cross section for typical mesospheric clouds. Can this be motivated better? Otherwise a larger range may be appropriate from n = 2 (geometrical optics limit) to n = 6 (Rayleigh limit). The reference to Hultgren and Gumbel (2014) has also been mentioned above. This reference is interesting even here as it discusses ideas underlying the relationship between cloud brightness and ice (including e.g. r^n dependence of the scattering coefficient, dependence on particle size distribution) that are also discussed in the current paper.

Figure 1: In order to better understand the behavior of the model data, it would be instructive for the reader to see the data points all the way down to the zero point (small albedo, small IWC). I do not see a reason not to show these points.

Figure 2: To avoid confusion, I suggest to mention the units of the contour lines (g km-2) in the figure caption.

Figure 11: In the two plots, there is an obvious lower limit to the data points (in terms of a straight line nearly parallel to the red line). I (and possibly other readers) do not understand why there is such a well-defined lower limit. Please add an explanation.

[Figure]

---

## Author Comment (AC1) · 2 Feb 2019

Thomas et al. Anonymous Referee #1 General comments: This manuscript deals with an alternative approach to retrieve the ice water content (IWC) of PMCs from satellite-borne UV-backscatter observations with the CIPS instrument on the AIM satellite. Due to orbital drifts CIPS is since 2016 operated in a different way and the original approach to retrieve IWC is not possible any more for a large part of the measurements. The novel approach estimates IWC from

backscatter measurements at a single scattering angle. Overall, the approach works well. The paper is generally well written and is suitable for publication in AMT. I ask the authors to consider the comments listed below. In my opinion the paper should be accepted, once the (mainly minor) issues listed below were addressed.

Specific comments and responses: Our responses are in italics. When we respond "OK", it means that we are following your recommendation. We also include a revised pdf document with our tracked changes, where all additions are in red type. Line numbers are referred to "new" if they refer to the new document, which has been reformatted.

Line 16: "265nm" -> "265 nm"OK

Line 114: "..it is necessary to derive scattered radiances from the same algorithm used by CIPS." I don't really understand this statement "derive scattered radiances from the same algorithm used by CIPS." CIPS measures scattered radiances and the algorithms are used to infer physical properties of PMCs, right? I guess I'm missing a point? No, we were not clear. We replaced the sentences (lines 113-117) , (now 91-94) with the following: "Since the SOFIE technique uses near-IR solar extinction in ice-water absorption bands, the primary measurement is ice water content. As shown in Sec 2.2, we reversed the process, to derive radiances from IWC, and then applied the same regression method to the results." [I suggest an alternative wording for the last sentence (and note the change from section 2.2 to 2.3)]: "As shown in Sec 2.3, we inverted the retrieved SOFIE IWC to derive the equivalent 265-nm albedo, and then applied the regression method described above to the results." Line 138: "the monochromatic scattering cross-section" It actually is the "differential" scattering cross section. Perhaps this can/should be added. OK. Line 140: "number density of ice particles" Perhaps better "column density"?

in new line 107 we changed "number density" to "column density".

Equation (2): I suggest using a slightly different symbol for the albedo than in equation

(1). The left hand sides of equation (1) and (2) are the same, but the right hand sides differ.

Agreed. We placed the letter "m" as a superscript, and identified it as such in the previous sentence (new line 117).

Line 167: add space before "denotes". OK

Line 209: I don't fully understand the meaning of this sentence. Is the microphysical model a 1-D model?

No. It is fully 3-D.

"(2) radiance and IWC may be calculated accurately, so that effects of cloud inhomogeneity are absent;"

We added the following sentences after (new) line 172." With regard to the latter point, we describe in more detail the model calculations. The model grid is 4o in latitude, 5o in longitude, and variable in the vertical. Ice particles of varying sizes fill many of these cells, but the density of particles within each cell is, by definition, constant. For a given model cloud, the integration is made through a vertical 'stack' of all ice-filled cells generated in a given computer run, and within each particle size grid. The total radiance is the sum of contributions from the size range 20 to 150 nm. The observation angles are always assumed to be zero, in other 175 words, the integration is performed in the vertical only. Thus cloud 'boundaries' in the horizontal plane are not an issue. This contrasts with real heterogeneous clouds where these approximations would not hold." As you will note in the response to Reviewer 2, we added two more clarifying statements.

Line 216: "thick.)." -> "thick.)."

Same line: I suggest adding "resolution" to read "The model height grid resolution is variable, being highest. ", otherwise "being highest" doesn't make sense.

We changed the sentence to read (now line 186) "The model height grid is variable, so that the smallest layer thickness is 0.26 km, which resolves the narrow ice layers" ... Line 220 and Figure 1: I'm wondering how the linear regressions are actually done. Is it a single linear fit of y vs. x, or do you fit both y vs. x and x vs. y and determine an average slope and offset? Alternatively, there are routines taking both differences in x and y direction into account, when minimizing chi-square. There may be a large difference between fitting y vs. x or x vs. y.

This is an interesting point. However it is largely a mathematical issue, irrelevant to the desired objective. The albedo is the measured quantity, and IWC is the derived quantity - the albedo is the natural independent variable. Certainly, there are circumstances where the opposite is true, when IWC is the measured variable (as in the case of SOFIE) and one desires to know the albedo- (however one would also have to specify the scattering angle in that case). As the reviewer notes, in this case the fitting process would have to be x vs y, but our regression results (y vs. x) apply to the question being asked.

Line 227: "In fact, we found that the linear relationship breaks down for very small albedo," The reason is probably, that particle populations with really small particles (< 10 nm) have a non-zero IWC, but the albedo is essentially zero, right?

Yes, we would agree with that. In line (new) 199, We replaced this phrase with "below the detection threshold of CIPS and SBUV, and are a result of the very faint small particles."

Figure 1: Both the IWC and the albedo values have lower limits. Perhaps I missed it, but what is the reason for this lower limit. For IWC the limit seems to be 20 g/km2.

"The lower limit for the albedo (SA=90o) is 1G, which is the detection limit of the CIPS experiment." We added this comment to the figure caption. . Figure 2: The caption should clarify that the displayed error is a relative error given in Line 216:

We added the word "relative" to the word 'error' in figure caption 2. We added further clarification by adding : "The errors are relative to the model values"

Closing parenthesis missing after "Rapp and Thomas (2006)". OK

Line new 114: "DIFFERENTIAL scattering cross-section"? OK

Line 289: "The mean difference is -13%. Looking at the Figure, the mean difference seems to be larger.

-13% is an unweighted mean over all bins. If we had weighted the error according to the number of points in each bin, the error would be larger, and positive. We added the sentence (now line 243)" Assuming the microphysical model is accurate, the accuracy of the CIPS UV measurements ranges from over +100% for very small albedo to -60% for high albedos. We emphasize that this is not an AIR result, but is an attempt to assess how particles that are too small to be visible to UV measurements affect the accuracy of the CIPS IWC results."

Also, what is the reason for the characteristic variation of the difference with increasing albedo? Is there a simple explanation?

We elaborate further in the Conclusion section (lines 353 -356): "Assuming the microphysical model is accurate, the accuracy of UV measurements ranges from over +100% for very small albedo to -60% for high albedos. The overall accuracy of IWC (averaging over all albedo bins) is -13±17%. The CIPS algorithm overestimates the small-particle population (20-30 nm) as a result of the Gaussian [this change should also be made in the paper] approximation when the mean particle size is small, and the opposite is true when the mean size is large. Distinct from the more fundamental errors due to the invisibility of very small ice particles and the Gaussian approximation, we also estimated the errors in the AIR approximation, relative to the AIM SOFIE data which apply to larger values of IWC than the model. For the dimmer and more frequent clouds, Fig. 2 shows that the error in ensemble averages is of the order of 5%.

[Figure]

Line 289: " .. subset .. have" -> ".. subset .. has" OK

Line 303: "As previously shown, the AIR data applies to the ice mass of 'UV-visible' clouds, not to their total ice mass." I don't really understand this statement. What is the meaning of "UV-visible clouds"? Was this shown in the current paper? If not, please provide a reference.

In line 112 we added the sentence "As shown by Rapp and Thomas (2006), particles with sizes < 20 nm are not detectable by UV measurements because of their small cross-section values – hence we refer to 'UV-visible clouds'."

Caption Figure 4, line 3: "SA = 40 deg" should read "SA = 50 deg". OK.

Line 366: "AIR overestimates IWC by up to 15% AIR may also underestimate IWC, right? I also think that the overestimation may be much larger than 15%.

See the above additions where we address these issues in more detail (line 243): "Assuming the microphysical model is accurate, the accuracy of the CIPS UV measurements ranges from over +100% for very small albedo to -60% for high albedos. We emphasize that this is not an AIR result, but is an attempt to assess how particles that are too small to be visible to UV measurements affect the accuracy of the CIPS IWC results."

Line 367: "SA = 100 deg" -> "SA = 110 deg" OK

Line 375: "The accuracy of the average IWC results was estimated by removing half the data from an entire season and comparing the two results. " I'm not sure, I fully understand this statement. AIR is applied to SBUV data and then you split the data set in half and compare the results. How does this allow you to estimate the accuracy of the "average IWC results"? I guess I'm missing the point here?

We added (line 101) the following sentence to emphasize that the errors refer to systematic differences between several divided data sets; "For a highly-populated region (more than 1000 clouds per season at latitudes higher than 70°), the differences in

IWC ranged between $\pm 3 - 5$ g-km-2, thus can be considered typical systematic errors. For a less-populated region ($50° - 64°$ latitude) where there were many fewer clouds (<50), the differences were larger, $\pm 5 - 10$ g-km-2."

Next sentence: You report on "changes" of $\pm 3$ - 5 g km2 etc., but were there any systematic differences? These values were the systematic differences between pairs of divided data sets.

We added the word "systematic" to $\pm 3$ - 5 g / km2. We specifically added the phrase "systematic" errors. See above.

---

## Author Comment (AC2) · 2 Feb 2019

Thomas et al. Anonymous Referee #2 This manuscript describes the retrieval of the ice water content (IWC) of mesospheric clouds based on measuring cloud albedo. This retrieval method has been developed with particular focus on AIM/CIPS since changes in orbit have made the original IWC retrieval based on phase function analysis impossible. However, the method is also applicable to other mesospheric datasets like SBUV. This gives the method potential

importance for inter-comparisons of mesospheric datasets and for the analysis of long-term variability. The approach is straight-forward, and the results are convincing. I recommend the manuscript for publication after some minor revisions. I mainly would like to see a number of clarifications.

New and old text are in "".

One issue I would like the authors to discuss more: Why is the AIR method not even better than described in the manuscript? The AIR method is based on finding a linear relationship between ice water content and cloud albedo. For typical mesospheric clouds, one can argue for such a relationship even on a theoretical basis, as long as scattering coefficients depend on the particle size to a power in the vicinity of 3. The authors show that the AIR method works well on a statistical basis. However, the AIR results presented in the manuscript show much scatter in the relationship between albedo and IWC, and the authors point out in several places that we cannot expect AIR to work for individual IWC retrievals. I would like to see more discussion on why the method is "not better than this". A reference is e.g. Hultgren and Gumbel (J. Geophys. Res., 119, 14129-14143, 2014). That paper shows many examples of a close relationship between cloud scattering coefficient and ice mass density, which works well even for the altitude-dependent quantities, not only for the column-integrated quantities considered in the current paper. Can I dare the authors to make a more quantitative statement: Can we take the AIR results for real and apply the method to individual retrievals, by providing a suitable statement about the error bar of such individual AIR IWC retrievals? How large (in percent) would such an error bar be?

Apart from the question of why one would be interested in only one cloud measurement, we have responded by calculating the overall percent error in single "measurements" (model simulations) of albedo, given the scattering angle. We looked at all albedos>1G and SA=90o. The distribution of AIR errors is given below. The std deviation of the Gaussian fit is 19%. For most applications this is probably too large an error to be useful. The dispersion of particle sizes leads to a distribution that is quasi-

random. This means that averaging will improve the mean by the square root of the number n of measurements. We separated the data into sets of 100 and 500 randomly chosen iwc-albedo pairs, and calculated the % errors. The errors in the means of the AIR distributions [sigma/sqrt(n)] were 2.16% and 0.9%; these are equivalent to a standard error of ∼19% for a random data set. These numbers will vary with scattering angle, and with specific albedo values, but this provides an example of how AIR would work with real data, on a time interval when only a few hundred measurements are available, e.g. one month.

We added the following sentence in line 225 (now line 225 ): "For the conditions in Fig. 1(c), the mean error of AIR for a single model simulation is 19%. The error can be reduced substantially by averaging. For example, for 100 measurements, the AIR error is only 2%. Figure 1 also shows…." We also added the reference Hultgren and Gumbel to line 130. See Supplemental Figure 1:"singlemeasurementerrors.jpeg

Section 2 (Theoretical basis): The major result of this study is that IWC is linearly related to cloud albedo. Therefore, the description in line 156-160 is confusing. In line 156-157, the authors refer to "the results of this study that IWC is linearly related to the column density of ice particles". The fact that IWC is linearly related to the column density of ice particles is somehow trivial (although dependent on the details of the particle population). In line 158-159, it is also stated that "As pointed out by Englert and Stevens (2007), such a relationship exists for certain SA values..." However, the relevant finding by Englert and Stevens is about the relationship between IWC and albedo. I therefore suspect that this paragraph should be about the relationship between IWC and albedo, not between IWC and column density. Please clarify and reformulate. We agree that we should have stated albedo rather than column density. We changed the sentence from "Anticipating the results of this study that IWC is linearly related to the column density of ice particles," to "Anticipating the results of this study that IWC is linearly related to cloud albedo, .." Section 2.1 (Model results): Please describe in more detailed the processes included in the model simulations and the resulting variability

in mesospheric clouds. Is gravity wave activity included in the simulations? Does the cloud database include multiple layer clouds or other conditions that may lead to clouds deviating from a straight growth/sedimentation scenario?

In response to reviewer 1, we added more detail on the model calculations (lines171-onward). In response to your specific request, we added the sentence (now line 177):" The model contains variability due to waves of various sorts, including tides and gravity waves. However, it does not capture all known details of PMC, such as double layers. Since we are dealing with integrated quantities, this should not be an important issue. Furthermore, we don't place full reliance on the model, which is why we also use two independent data sets." Section 2.2 (AIR results from CIPS): Figure 8 shows the AIR method applied to the CIPS from the Northern Hemisphere 2011. When it comes to demonstrating that the AIR method works, this choice of season is unfortunate. CIPS data from the years 2010-2013 has been used in the regression analysis to "train" the model. When subsequently investigating the ability of the model to retrieve IWC, a season should be chosen that has not already been used to train the model. I suggest to choose another season. Since we 'trained' the AIR method to literally hundreds of thousands of individual cloud data, showing how it works for a single day does not detract from illustrating its usefulness.. Some other details: Line 84: The notation "meteor 'smoke' nucleation" may be misleading. It is better to write "nucleation on meteoric 'smoke'".

Agreed: We replace this phrase by "The processes treated by the model include nucle-ation on meteor 'smoke' particles,.." Line 274: To avoid confusion, please clarify what is meant by "mean ice particle volume evaluated at rm", i.e. make clear that you refer to an integration over the Gaussian particle size distribution.

Agreed. We replaced the sentence in (now) line 231 with "V denotes the ice particle volume, averaged over the Gaussian distribution with a mean particle radius value rm." Line 287: Clarify that by "simulated CIPS retrieved IWC" you mean the AIR result. We apologize for this misconception. We added the following sentence (line 243): "We

emphasize that this is not an AIR result, but is an attempt to assess how particles that are too small to be visible to UV measurements affect the accuracy of the CIPS IWC results." Line 394: The authors refer to n = 3-5 as typical exponents for the size-dependence (sigma âĹij rËĘn) of the scattering cross section for typical mesospheric clouds. Can this be motivated better? Otherwise a larger range may be appropriate from n = 2 (geometrical optics limit) to n = 6 (Rayleigh limit). The reference to Hultgren and Gumbel (2014) has also been mentioned above. This reference is interesting even here as it discusses ideas underlying the relationship between cloud brightness and ice (including e.g. rËĘn dependence of the scattering coefficient, dependence on particle size distribution) that are also discussed in the current paper.

See below for our response to why AIR works as well as it does. Figure 1: In order to better understand the behavior of the model data, it would be instructive for the reader to see the data points all the way down to the zero point (small albedo, small IWC). I do not see a reason not to show these points.

For your curiosity, this supplemental figure 2 shows the results down to very small albedo. This information is not relevant to the CIPS data, which has a detection limit of 1G. The behavior of the relationship is different, with a different slope and more dispersion. We believe that showing this behavior detracts from our message. The blue dots show results from effective particle sizes < 20nm. Red dots show results for reff between 20 & 30 nm and the black dots show the contributions from larger sizes. The light blue line is the AIR result, which is curved because of the log-log scale.

See Supplemental Figure 2, "IWCvsfaintalbedo.jpeg'

Figure 2: To avoid confusion, I suggest to mention the units of the contour lines (g km-2) in the figure caption.

Again, we apologize for the misconception. We added the sentence: "Contour lines are labelled as percent errors relative to the accurate model values."

[Figure]

Figure 11: In the two plots, there is an obvious lower limit to the data points (in terms of a straight line nearly parallel to the red line). I (and possibly other readers) do not understand why there is such a well-defined lower limit. Please add an explanation.

This was a very helpful suggestion, and we looked further into this matter. The ratio of albedo to IWC shows a clearly defined lower limit to this ratio, which is nearly independent of the effective particle size at least for the larger particles.. This behavior is due to the fact that the ratio of IWC/albedo (which is independent of column density) asymptotes to a straight line for the larger values of particle size (30 <reff<50 nm) which are those responsible for PMC. Below is a plot of this ratio versus effective radius, showing this behavior, for SA=90 deg. The constant ratio at large reff is due to the fact that the r-dependence of the cross-section in this range is r3, and cancels the IWC dependence on r3. The same behavior occurs for the other scattering angles, but the asymptotes are different. The near-constancy of the ratios means that there will be distinct lower limits to the regressions of IWC vs Albedo, which slant upwards in the plots due to the linear variation of albedo on column density. This is evident in Figure 1 where the lower boundary of the scatter is quite linear. These points refer to the largest particles in the population. The other reason for the sharpness of the lower boundary of the ratio is that the largest particles are the ones responsible, and these have a very steep fall-off in the size distributions.

See Supplemental Fig. 3, "RatioIWCtoAlb.jpeg

We added the discussion of this issue in Sec. 3, Effects of Mean Particle Size, beginning on line 329, "The AIR approximation is based on the notion that particle size effects can be ignored in retrieving IWC from albedo measurements. That is, they contribute in a sense to the 'noise' of the measurement, which can be minimized by averaging. In fact, the particle size (or more accurately, the term ðÍŚ§3) is a principal 'driver' of < ðÍŘijðÍŚŁðÍŘű > itself, so it is not obvious that particle size effects play a minor role in deriving IWC. The dependence of albedo on column density adequately captures this part of the variability (albedo is strictly linear in column density). The

AIR slope term is $\sim r^3/\sigma\_lambda(r,phi)$ averaged over a distribution of particle sizes, r. The size dependence of the cross-section varies as a power of r, within two limits, the geometric-optics limit, r2, and the small-particle (Rayleigh) limit, r6. In the intermediate and realistic conditions of PMC, the exponent has an intermediate value. Fortunately, there is a "sweet spot" (or better, a 'sweet region' of the r-domain) in which the r-dependence of sigma_lambda is $\sim r^3$, so that the slope term is constant (for fixed SA). This behavior occurs for all relevant values of SA, and for the albedo values typical of CIPS. It accounts mainly for the effectiveness of the AIR method. The other aspect favorable to AIR is the steep fall-off of the particle size distribution at the largest sizes, which contributes to the sharpness of the lower boundaries in the spread of points in Fig. 1."

[Figure]

**Fig. 1.** Supplemetal fig 1

...

SD/WACCM-ICE Model,SA=90

☒ IDL 4

IWC (gm km$^{-2}$)

Albedo (90°), G

**Fig. 2.** supplemental fig 2

**Fig. 3.** supplemental fig 3